# Exploring Strategies for Developing Enabling Environments for People with Chronic Heart Disease: An Ethnographic Study Protocol

**DOI:** 10.3390/ijerph20032680

**Published:** 2023-02-02

**Authors:** Valérie Loizeau, Kelley Kilpatrick, Dominique Pougheon Bertrand, Monique Rothan-Tondeur

**Affiliations:** 1Centre Hospitalier Poissy Saint Germain-GHT Yvelines Nord, 78300 Poissy, France; 2Nursing Sciences Research Chair, Laboratory Education and Health Promotion (LEPS UR 3412), UFR SMBH, Université Sorbonne Paris Nord, 1 Rue de Chablis, 93000 Bobigny, France; 3Susan E. French Chair in Nursing Research and Innovative Practice, Ingram School of Nursing, Faculty of Medicine, McGill University, Montreal, QC H3A 2M7, Canada; 4Laboratory Education and Health Promotion (LEPS UR 3412), UFR SMBH, Université Sorbonne Paris Nord, 1 Rue de Chablis, 93000 Bobigny, France; 5AP HP, Nursing Sciences Research Chair, 55 Boulevard Diderot, 75012 Paris, France

**Keywords:** chronic illness, enabling environment, empowerment, cardiovascular disease

## Abstract

The impact of chronic diseases on people’s daily lives and the exponential number of people affected is a major public health issue. The consequences on individuals and their families is significant, particularly in terms of quality of life. In the literature, this phenomenon is well described in terms of care policy and cost. Although there is a link between a supportive environment and empowerment, there is little literature describing a supportive environment and the daily lives of people living with cardiovascular disease. The objectives of this study are to identify the strategies people use to develop an enabling environment. It will be a qualitative ethnographic study that will address both human behavior and the notion of culture in a broad sense. In the context of this study, an orientation towards critical ethnography will be considered for its particular interest in vulnerable people and in the power relations that may exist in the socio-cultural system. Data will be collected directly in people’s homes through observations and interviews with 10 people with cardiovascular disease. For each person, the data collection will take place over three days and will represent approximately 210 h of observation. This protocol was registered in the Research Register on 30 June 2021 and its number is 6933. This study will explore strategies for developing an enabling environment for people living with heart disease and eventually provide recommendations for nursing practices in terms of support.

## 1. Introduction

While 3 out of 5 deaths worldwide are attributed to 4 diseases (cardiovascular disease, cancer, diabetes and chronic lung disease) [1], chronic diseases, referred to by the WHO (World Health Organization) as non-communicable diseases, cause death every year [2]. As the leading cause of death in the world for people under 70, cardiovascular diseases are disabling for both individuals and their families [3]. Among them, chronic heart failure affects at least 26 million people worldwide [4]. The median rate for people over 60 years of age is 11.8% and is expected to increase over the next 20 years [3]. In 2012, this disease was responsible for an expenditure of USD 31 million in USA, i.e., 10% of the total health expenditure [5]. At the same time, the health of the world’s population has improved over the past 30 years, with the rate of disability-adjusted life years remaining stable. Disability caused by chronic diseases and an ageing population also require management policies [6]. Chronic diseases are seen as a burden. They lead to an altered quality of life. The person has to revisit his or her life and live with the disease [1]. The notion of living with a chronic disease has changed in recent years from quantity to quality. The individual’s perception of quality of life requires a self-assessment that depends on internal and external factors specific to the individual. For example, in the case of diabetes, factors include adherence to treatment, depression and marital status [7]. Similarly, when living with heart disease, depression, age and gender have a negative impact on the quality of life [8]. This notion of quality of life seems to be linked to the environment in which the person with the disease lives. The environment is often linked to noise, green spaces and air quality, with pollution being a major factor in the management of respiratory diseases [9,10]. However, the notion of environment includes individual factors such as age, illness and factors linked to the social environment [11].

Furthermore, a favorable environment allows people to develop their empowerment, defined by Gibson as a process of increasing a person’s ability to solve their own problems and mobilize the necessary resources [12]. This environment, described by some authors as enabling, allows resources to be adapted to the person so that he or she can develop empowerment [13].

The availability of resources in the environment is not in itself sufficient to provide for the autonomy of people living with a chronic disease. Indeed, the environment must meet favorable conditions such as support and commitment between health care professionals and people, allowing in particular an improvement in their knowledge [14]. There is a link between low health literacy and disease self-management. Indeed, people have difficulties in processing information. In cardiovascular diseases, adherence to treatment is complicated, and the diet is poorly followed, which increases the number of hospitalizations [15]. In addition, the capability approach described by the Indian economist Amartya Sen is a framework for assessing individual well-being. It focuses on the strengths and rights that enable people to seize opportunities to act [11]. This approach integrates fundamental concepts such as functioning, and what the person can achieve or be. For this, conversion factors are needed, which are specific to the individual, and relate to the environment, including the living environment [16]. Thus, it is necessary to distinguish between what the person achieves and what is possible for him/her to achieve [17].

The aim of this study is to gather data on the living environment of people in relation to the development of a supportive environment. To do this, the researcher will observe the organization of people’s environment in relation to the management of cardiovascular disease. This will enable professionals to take an interest in the real expectations of people during follow-up consultations. In addition, they will be able to propose innovative solutions by adjusting certain care practices.

## 2. Materials and Methods

### 2.1. Ethical Aspects

This protocol was approved by the ethics committee CER U-Paris N° 2021-25 on 4 May 2021. Participants will be informed by the principal investigator, given an information letter and consent will be obtained.

### 2.2. Design

This study has two objectives: to identify the strategies for setting up an enabling environment for people in their daily lives at home and to describe the factors that encourage and hinder its development. In order to answer these questions, we will carry out a qualitative ethnographic study which will allow us to look at human behavior and the notion of culture in the broadest sense (values, and beliefs of the same group). The choice of this method is inherent to the reality of the person with the disease in his or her daily life, and the observation of his or her frame of reference to develop his or her environment. Traditional ethnography varies according to the nature of the phenomenon studied, and so offers other ethnographic approaches. In the context of this study, an orientation towards critical ethnography [18] will be chosen because of its focus on vulnerable people and on the power relations that may exist in the socio-cultural system. In addition, this variant of traditional ethnography will make it possible to observe the situation in the present, to imagine what could happen and, in particular, to ‘stimulate the actors’ power to act’ [19]. Depending on the results of the study, the discussion part will focus on this notion of empowerment and the power relationship between the person and the care professionals in the person’s environment.

### 2.3. Settings and Participants

A study will be conducted among people with cardiovascular disease followed in a public hospital selected on the basis of the geographical proximity of the researcher. These people are usually followed by cardiologists working in this public hospital providing outpatient consultations. Non-probability purposive sampling will be used for selection and re-recruitment of the population consulted [20]. In ethnography, Leininger states that 5–6 people are sufficient to achieve the depth of data and maintain scientific rigor [21]

### 2.4. Data Collection

The investigator will visit people’s homes over three days. In order to collect data for this study, two methods are envisaged: observations including informal conversations and semi-directed interviews. The research will take place in the patients’ homes over the course of three days per patient.

### 2.5. Situational Observations

The objective of these observations is to collect data that will allow us to describe the manner and strategies used by people to develop their environment in everyday situations. These observations will be carried out by the principal researcher of the study (PhD), who is a nurse by profession. Without a preconceived stance, an observation grid will make it possible to target what is happening in the environment, and to look for the event in its entirety from different points of view [22]. In addition, the observation grid makes it possible to ‘centre the researcher’s gaze’ [23]. The idea is also that the event can be observed in its entirety from different perspectives, as well as permitting the observation of new phenomena. This observation grid is constructed on the basis of two elements: the work of the laboratory to which the research team belongs, and the results of the integrative review by Loizeau [14]. This grid includes several elements related to the description of the person’s environment: the support of the entourage, the way in which the person improves his or her knowledge, the evaluation of his or her needs, the listening to his or her concerns, the relationship with the professionals with whom he or she works and the organization of care in the place of living.

During these observations, informal conversations will enrich the data and draw out new elements. These initially passive observations could become participatory because of the origin of the investigator’s nursing profession, undoubtedly allowing her to better understand certain elements of an individual’s behavior, as well as justifying her presence at key moments in their lives.

### 2.6. The Interviews

These in-depth ethnographic interviews will allow a better understanding of the development of this environment [24,25]. They will be carried out as an extension of the observations and according to the availability of the subjects. In the context of this study, the interview will take place on the third day of the investigator’s presence in the living environment. A somewhat flexible interview guide will be used. The themes will also be related to the management of the disease on a daily basis, collaboration with professionals, consideration of expectations and concerns and the support network.

The aim of these interviews is to explore in greater depth certain points observed or not, based on the elements noted in the observation grid and during informal conversations. The time allotted for these ethnographic interviews will not exceed one hour.

### 2.7. Classification and Transcription of Data

Before starting the analysis, the data will be classified and transcribed. A summary table of people’s characteristics will be produced and anonymized. This will include data on age, pathology, marital and professional status, description of place of living and family. The time spent observing and interviewing people will also be noted.

The observations will be transcribed in Word using the digital logbook, which will save time. The interviews will be integrated and processed using Atlas Ti (V6) software adapted for qualitative research. The data from the observations and interviews will be combined in a single file for each person.

### 2.8. Analysis

The analysis will be conducted in several stages, with three researchers planned. The principal investigator will gather the data collected through observations, interviews and informal conversations in Atlas Ti software. Initially, there will be several readings of the data and then simple coding. This step will be carried out by two researchers, with a third validating in case of dispute. In the second stage, the units of meaning will be taken for a four-phase analysis according to the method proposed by Spradley [25]. This method makes it possible to highlight the meaning that people give to their environment. The first phase consists of analyzing the domain, i.e., grouping together the codes previously highlighted and having at least one characteristic in common, with each code having a precise meaning. We thus will obtain a list of domains representing behaviors in a particular context [19]. The second phase is the taxonomic analysis, which will allow us to organize the codes associated with the same domain. The result is a classification of the meanings of the behaviors and the elements of the context. The third phase consists of the component analysis, which deepens both the domain analysis and the taxonomic analysis. The idea is to highlight the distinctive codes. The fourth phase is the analysis of the themes, which consists in reformulating them to make sense of people’s behaviors. The final analysis will highlight the subjects’ strategies for managing their environment and the factors that encourage them. The validation of this analysis will also be monitored by a second researcher from the Nursing Research Chair.

### 2.9. Potential Limitation and Bias

There are no other risks inherent in this research apart from possible psychological risks that may be induced during the three home visits. These risks can be prevented by adopting as neutral a stance as possible, particularly through the words spoken during informal discussions and facial expressions during the interviews, and by establishing a climate of trust between the person (or those around them) and the investigator [26]. Moreover, the investigator’s position is all the more important, as her original profession is that of a nurse, and she may be particularly solicited by the subject during the observation and informal conversation. The subject may also confide in her at particular times of the day, particularly in relation to the management of their illness [19].

Similarly, the limitation of recruitment bias is envisaged: the subjects entering the study will be chosen according to their pathology and the seriousness of this in the management of their daily life. They will therefore not be recruited according to their social and cultural environment. The place of living, which is an element of a person’s environment, will not be chosen by the investigator nor anticipated during recruitment. The recruitment process chosen during the consultation process will thus make it possible to minimize this bias.

To try to reduce the Hawthorne effect in our study, the data collection is multiple, consisting of observations, informal discussions and interviews. Studies using covert observation can help avoid the Hawthorne effect, although even knowledge of study participation itself is considered to have the potential to induce a Hawthorne effect [27]. A paragraph on the individual’s reactions will be included in the final study document [28]. These three modalities carried out at different times of the day and over three days should allow people to feel more at ease and reduce the sense of being part of a study. It is planned to make people feel at ease from the start of the study by repeated explanations and by accommodating the individual’s schedules and activities.

In order to limit the risks linked to COVID-19, precautions will be taken with regard to both barrier gestures and the health status of the participants (surgical mask changed every 4 h, hydro-alcoholic gel, cleaning of equipment before entering the home). The days of attendance will be scheduled according to the health status of the person and their family. In addition, the investigator will be vaccinated against COVID-19 during her involvement with the subjects in their homes.

## 3. Discussion

The number of people affected by one or more chronic diseases, particularly cardiovascular diseases, is continuously increasing in France and throughout the world, as is the impact in terms of quality of life and cost to society. Faced with this public health problem, one of the responses of health policies is based on the development of empowerment, in other words, the autonomy of individuals [29]. It seems obvious to leave people free to make their own choices and decisions in the daily management of their illness [30]. However, there is a risk of over-empowerment for some by allowing them to manage their disease in its entirety [31].

In fact, the results of this innovative ethnographic study will be discussed in terms of support in relation to the interventions proposed by the professionals according to the needs and expectations of those living with chronic disease. Moreover, the data from the observations in these people’s living environment can put into perspective the resources proposed in the framework of the care pathways and their actual implementation in a person’s daily life.

In addition, this protocol is part of a research program questioning the enabling environment in the context of chronic illness in relation to the activities of advanced practice nurses. The implementation in France of advanced practice in the care pathways will be discussed in light of people’s environments.

## Data Availability

Not Applicable.

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
