# Peer review of "Exploring Strategies for Developing Enabling Environments for People with Chronic Heart Disease: An Ethnographic Study Protocol"

_ijerph, 2023, doi:10.3390/ijerph20032680_

Round 1
Reviewer 1 Report
The aim of this study protol is to identify strategies for setting up an enabling environment for people in their daily lives, and to describe the factors that promote or hinder the development of this environment. The protocol is well designed while the writing needs to be improved.
The Abstract and Introduction part is not well-written. Information is missing here and there. Very long sentences make it hard to understand what the authors mean. The discussion part is clear and easy to understand.
Line18: In the literature, this phenomenon is well described in terms of care policy, burden, pathways, and cost.
I didn’t find these details in this paper, please add more details according to these four aspects.
Line 23: and to describe the factors that promote or hinder the development of this environment.
Please list some potential factors that may hinder the development of enabling environment in introduction if there were related studies before.
Line 37-39 While 3 out of 5 deaths worldwide are attributed to 4 diseases [1], chronic diseases, named by the WHO (World Health Organisation) as non-communicable diseases, reach and kill every year making them the leading cause of death in the world.
What are 4 diseases? Reach and kill sounds not clear, reach which extend? Please rephrase this sentence to make it easier to understand.
Line44-46: Considered as a burden by those affected, chronic diseases lead to an altered quality of life, an inability to work, costs and a change in social life.
Affected+ noun. (Increased) cost. Please rephrase this sentence.
2.8 Analysis
Line 189: Spradley suggest XXX :
Please add a reference to this sentence
Line 200. Authors discussed the nurse position is the most important, please clarify what consequence is going to show if the nurse is not objective and how to improve this.
Reference:
Reference 5 need to be modified.
Author Response
Thank you for your attention to our submission. We have reviewed your comments points by points.
Kinds regards
Thank you for your attention to our submission. We have reviewed your comments point by point. We would like to point out that the manuscript was proofread by an English language editor. We believe that this is also related to the rather new concepts. We hope that our responses will lead to the publication of this article.
Point 1: The aim of this study protol is to identify strategies for setting up an enabling environment for people in their daily lives, and to describe the factors that promote or hinder the development of this environment. The protocol is well designed while the writing needs to be improved.|The Abstract and Introduction part is not well-written. Information is missing here and there. Very long sentences make it hard to understand what the authors mean. The discussion part is clear and easy to understand.
Response 1: Thank you for your comment, the abstract and introduction have been reworded. Put directly into the document
Point 2: Line18: In the literature, this phenomenon is well described in terms of care policy, burden, pathways, and cost. I didn’t find these details in this paper, please add more details according to these four aspects.
Response 2: the sentence has been modified, in the introduction we find the reference notably in terms of cost
Point 3: Line 23: and to describe the factors that promote or hinder the development of this environment. Please list some potential factors that may hinder the development of enabling environment in introduction if there were related studies before.
Response 3: the sentence has been changed, it was badly formulated
Point 4: Line 37-39 While 3 out of 5 deaths worldwide are attributed to 4 diseases [1], chronic diseases, named by the WHO (World Health Organisation) as non-communicable diseases, reach and kill every year making them the leading cause of death in the world. What are 4 diseases? Reach and kill sounds not clear, reach which extend? Please rephrase this sentence to make it easier to understand.
Response 4: The 4 diseases have been added in the text (cardiovascular diseases, cancers, diabetes, chronic lung diseases). You are right, the sentence has been changed
Point 5: Line44-46: Considered as a burden by those affected, chronic diseases lead to an altered quality of life, an inability to work, costs and a change in social life. Affected+ noun. (Increased) cost. Please rephrase this sentence.
Response 5: thank you for your comment, we have reworded the sentence 2
Point 6: Line 189: Spradley suggest XXX :Please add a reference to this sentence
Response 6: We have redone the paragraph at the request of reviewer 2 to make it more readable. The Spradley reference appears earlier in the text
Point 7: Line 200. Authors discussed the nurse position is the most important, please clarify what consequence is going to show if the nurse is not objective and how to improve this.
Response 7: The nurse has been trained for observation as part of her studies and specialisation. She will make it clear to people that she is there for research and not for care. She has already conducted many interviews as part of her work
Point 8: Reference 5 need to be modified.
Response 8: Thank you for your attention the reference has been revised

Reviewer 2 Report
Abstract
Fails to mention chronic heart disease in to the background or in initial aims, despite being a focal point of the title and a keyword. Fails to link cardiovascular disease as a chronic disease or why this population was chosen.
Line 47
Should this be an emphasis on healthcare, rather than impact? Difficult to understand this sentence.
Line51
Do these variables have a negative or positive impact on quality of life? Unclear in current wording.
Line 54
Why is the link to respiratory pathologies relevant to this paper when the focus will be on cardiovascular disease?
Line 55
Mentions environment as a global notion but then puts emphasis on individualised factors in the same sentence.
Line 68
Mentions knowledge but this section would benefit from further exploration about health literacy in chronic disease and its impact on environment.
Introduction
Fails as a whole to clearly define the problem in relation to the projects aim and subsequent methodology. There are weak links between the burden of chronic disease, quality of life and the environment that require further review and exploration. More detail as to the burden of chronic disease both on the individual and economy, use of health services, and more specifically where cardiovascular disease sits within chronic disease in terms of incidence, prevalence and severity would make the need for the study more impactful. It would also add understanding as to why participants with cardiovascular disease were chosen for the study.
Line 107
This sentence would be better placed in either setting or data collection.
Section 2.3
This formatting is inappropriate and should be one paragraph or split into separate sections
Section 2.3
How has this sampling occurred, who made the initial identification, what potential biases are there, what criteria. What was the time period for recruitment, what kind of consultations – outpatient, public or private settings, etc. More detail is required. Eligibility criteria should be together (not separated by recruitment, which needs additional info as above)
Section 2.5
Could the observational grid be included as an example table or appendix?
Section 2.6
Similarly, could the interview guide be provided?
Section 2.8
Reformatting would significantly improve readability. Who is doing the readings, how many people, how will the coding being undertaken – e.g. all to develop initial and then come together, two rounds of this etc? A flowchart of the methodology may be beneficial also.
Line 198
What kind of training has the nurse had to prevent bias and improve neutrality during interviews and observations?
Line 204
Given the known overrepresentation of culturally and linguistically diverse and lower socioeconomic populations within more severe chronic disease presentations, how will this be accounted for?
Line 224
This is the first mention of cardiovascular disease, as per earlier comments, this population needs to be introduced from the start of the manuscript.
Discussion
Referencing is required in the discussion. (Line 226, 228, 229, 230).
Discussion
Should this be called the conclusion?
References
#8,9,15,20 – is there an updated reference available?
#17 – reference appears incomplete, nil year of publication
Author Response
Thank your for your review and comments. We have taken everything back as agreed.
Kinds regards
Thank you for your attention to our submission. We have reviewed your comments point by point. We would
like to point out that the manuscript was proofread by an English language editor. We believe that this is also
related to the rather new concepts. We hope that our responses will lead to the publication of this article.
Point 1 Fails to mention chronic heart disease in to the background or in initial aims, despite being a focal point of the title and a keyword. Fails to link cardiovascular disease as a chronic disease or why this population was chosen.
Response 1: Thank you for your comment, you are right, we will note why we chose cardiovascular diseases in the abstract.
here is the abstract reworked according to your comments :
The impact of chronic diseases on people's daily lives and the exponential number of people affected is a major public health issue. The consequences on individuals and their families is significant, particularly in terms of quality of life. In the literature, this phenomenon is well described in terms of care policy and cost. Although there is a link between a supportive environment and empowerment, there is little literature describing a supportive environment and the daily lives of people living with cardiovascular disease. The objectives of this study are to identify the strategies people use to develop an enabling environment. It will be a qualitative ethnographic study that will address both human behaviour and the notion of culture in a broad sense. In the context of this study, an orientation towards critical ethnography will be considered for its particular interest in vulnerable people and in the power relations that may exist in the socio-cultural system. Data will be collected directly in people's homes through observations and interviews with 10 people with cardiovascular
disease. For each person, the data collection will take place over three days and will represent approximately 210 hours of observation. This protocol was registered in the research register on 30/06/2021 and its number is 6933. This study will explore strategies for developing an enabling environment for people living with heart disease and eventually provide recommendations for nursing practice in terms of support.
Point 2: Line 47 Should this be an emphasis on healthcare, rather than impact? Difficult to understand this sentence.
Response 2: You are right, we are taking away the notion of health care.
We propose: The notion of living with a chronic disease has evolved in recent years from quantity to quality.
Point 3: Line51 Do these variables have a negative or positive impact on quality of life? Unclear in
current wording.
Response 3: you are right, we propose another formulation
Similarly, when living with heart disease, depression, age and gender have a negative impact on
quality of life
Point 4: Line 54 Why is the link to respiratory pathologies relevant to this paper when the focus will be on cardiovascular disease?
Response 4 : Thank you for your comment, we have made the link with the environment in a global way. Indeed, when we talk about the environment nowadays, we often think of air and pollution.
Point 5: Line 55 Mentions environment as a global notion but then puts emphasis on individualised factors in the same sentence.
Response 5: Thank you, indeed it is a problem of wording, we suggest : The notion of environment includes individual factors such as age, illness and factors linked to the social environment.
Point 6: Line 68 Mentions knowledge but this section would benefit from further exploration about health literacy in chronic disease and its impact on environment.
Response 6: Thank you for your comment, we have added this paragraph
“There is a link between low health literacy and disease self-management. Indeed, people have difficulties in processing information. In cardiovascular diseases, adherence to treatment is complicated, the diet is poorly followed, which increases the number of hospitalisations”
Point 7: Introduction : Fails as a whole to clearly define the problem in relation to the projects aim and subsequent methodology. There are weak links between the burden of chronic disease, quality of life and the environment that require further review and exploration. More detail as to the burden of chronic disease both on the individual and economy, use of health services, and more specifically where cardiovascular disease sits within chronic disease in terms of incidence, prevalence and severity would make the need for the study more impactful. It would also add understanding as to why participants with cardiovascular disease were chosen for the study.
Response 7: Thank you, the paragraph has been revised according to your comments
1. Introduction
While 3 out of 5 deaths worldwide are attributed to 4 diseases (cardiovascular disease, cancer, diabetes, chronic lung disease) [1], chronic diseases, referred to by the WHO (World Health Organization) as non-communicable diseases causes death every year. As the leading cause of death in the world for people under 70, cardiovascular diseases are disabling for both individuals and their families. Among them, chronic heart failure affects at least 26 million people worldwide (Ambrosy). The median rate for people over 60 years of age is 11.8% and is expected to increase over the next 20 years (Van Riet). In 2012, this disease was responsible for an expenditure of 31 million dollars in the United States, i.e. 10% of total health expenditure (Savaresse). At the same time, the health of the world's population has improved over the past 30 years, with the rate of disability-adjusted life years remaining stable. Disability caused by chronic diseas-es and an ageing population also require management policies [3]. Considered as a bur-den, chronic diseases lead to impairment of quality of life, inability to work, costs and change in social life [1]. The notion of living with a chronic disease has changed in re-cent years from quantity to quality. The individual's perception of quality of life requires a self-assessment that depends on internal and external factors specific to the individual. For
example, in the case of diabetes, factors include adherence to treatment, depression and marital status [4]. Similarly, when living with heart disease, depression, age and gender have a negative impact on quality of life [5]. This notion of quality of life seems to be linked to the environment in which the
person with the disease lives. The environment is often linked to noise, green spaces and air quality, with pollution being a major factor in the management of respiratory diseases [6,7]. However, the notion of environment in-cludes individual factors such as age, illness and factors linked to the social environment [8]. Also, a favourable environment allows people to develop their empowerment, defined by Gibson as a process of increasing a person's ability to solve their own problems and mobi-lise the necessary resources [9]. This environment, described by some authors as enabling, allows resources to be adapted to the person so that he or she can develop empowerment [10]. The availability of resources in the environment is not in itself sufficient to provide for the autonomy of people living with a chronic disease. Indeed, the environment must meet fa-vourable conditions
such as support and commitment between health care professionals and people, allowing in particular an improvement of their knowledge [11]. There is a link between low health literacy and disease self-management. Indeed, people have difficulties in processing information. In cardiovascular diseases, adherence to treatment is compli-cated, the diet is poorly followed, which
increases the number of hos-pitalizations (Cajita). Also, the capability approach described by the Indian economist Amartya Sen is a framework for assessing individual well-being. It focuses on the strengths and rights that enable people to seize opportunities to act [12]. This approach integrates
fundamental concepts such as functionings, what the person can achieve or be. For this, conversion factors are needed, which are specific to the individual, and relate to the environment, in-cluding the living environment [13]. Thus, it is necessary to distinguish between what the person achieves and what is possible for him/her to achieve [14].
The aim of this study is to gather data on the living environment of people in relation to the development of a supportive environment. To do this, the researcher will observe the organisation of people's environment in relation to the management of cardiovascular disease. This will enable professionals to take an interest in the real expectations of people during follow-up consultations. In addition, they will be able to propose innovative solu-tions by adjusting certain care practices.
Point 8: Line 107 This sentence would be better placed in either setting or data collection.
Response 8: you are right, the sentence has been added in the data collection part
Point 9: Section 2.3
This formatting is inappropriate and should be one paragraph or split into separate sections
Response 9: the paragraph has been modified according to the comments
Point 10: Section 2.5 Could the observational grid be included as an example table or appendix?
Response 10: Thank you for your comment, the grid will be attached as a table
Point 11: Section 2.6 Similarly, could the interview guide be provided?
Response 11: Thank you for your comment, the interview guide will be attached as a table
Point 12: Section 2.8 Reformatting would significantly improve readability. Who is doing the readings, how many people, how will the coding being undertaken – e.g. all to develop initial and then come together, two rounds of this etc? A flowchart of the methodology may be beneficial also.
Response 12: reformatting has taken place as requested by the reviewer
The analysis will be done in several stages, with three researchers planned. The lead researcher will collate the data collected through observations, interviews and informal conversations into the Atlas IT software. This allows the data to be organised. In the first stage there will be several readings of
the data and then a simple coding. This step is carried out by two researchers, with a third validating in case of dispute.
In the second stage, the units of meaning will be taken back for a four-phase analysis according to the method proposed by Spradley [22]. This method makes it possible to highlight the meaning that people give to their environment.
The first phase consists of analysing the domain, i.e. grouping together the codes previously highlighted and having at least one common characteristic, each code having a precise meaning. This produces a list of domains representing behaviour in a particular context. This first phase links the codes by grouping them into a domain and answering "could be related to" [16].
The second phase is the taxonomic analysis, which will allow the codes associated with the same domain to be organised. The result is a classification of the meanings of the behaviours and the elements of the context. The validity of all codes with the same relationship is checked.
The third phase consists of the component analysis, which deepens both the domain analysis and the taxonomic analysis. The idea is to highlight the distinctive codes.
The fourth phase is the analysis of the themes, which consists of their reformulation to give meaning to people's behaviour. It integrates the three previous phases.
The final analysis will highlight the subjects' strategies for managing their environment and the factors that encourage them. The validation of this analysis will also be controlled by a second researcher of the Chair of Nursing Research.
Point 13: Line 198 What kind of training has the nurse had to prevent bias and improve neutrality
during interviews and observations?
Response 13: first of all, the researcher is a trained nurse and is used to interviewing sick people. Moreover, because of COVID, the researcher kept his mask on, as the precautions provided for.
Point 14 : Line 204 Given the known overrepresentation of culturally and linguistically diverse and lower socioeconomic populations within more severe chronic disease presentations, how will this be accounted for?
Response 14: When selecting people, the inclusion criteria will not take this into account, ethically this would be questionable. People will be selected according to the order in which they arrive at the consultation and according to the inclusion criteria.
Point 15 : line 224 This is the first mention of cardiovascular disease, as per earlier comments, this population needs to be introduced from the start of the manuscript.
Response 15: we have taken your comment into account
Point 16: Discussion Referencing is required in the discussion. (Line 226, 228, 229, 230).
Response 16: we propose the following references
Point 17: Discussion Should this be called the conclusion?
Response 17: No, it is a plan, but it would be revised according to the results of the study
Point 18: References
Response 18: reference 8 concerns the leading author of capability theory
reference 9 is a conceptual analysis of empowerment that has not been done since
reference 15 refers to the leading author of critical ethnography
reference 20 the same for ethnography
référence 17 – reference appears incomplete, nil year of publication you are right, it is an oversight

Round 2
Reviewer 1 Report
The authors has improved the abstract and introduction part, it's well writen and easy to understand. More details were added and correct reference were added. The protocol is sceientifically sound and the experimental design is appropriate.
Reviewer 2 Report
Point 10: Section 2.5 Could the observational grid be included as an example table or appendix?
Response 10: Thank you for your comment, the grid will be attached as a table
Reviewer comment: This cannot be seen in the revised manuscript. Please provide for review.
Point 11: Section 2.6 Similarly, could the interview guide be provided?
Response 11: Thank you for your comment, the interview guide will be attached as a table
Reviewer comment: This cannot be seen in the revised manuscript. Please provide for review.
Point 13: Line 198 What kind of training has the nurse had to prevent bias and improve neutrality
during interviews and observations?
Response 13: first of all, the researcher is a trained nurse and is used to interviewing sick people. Moreover, because of COVID, the researcher kept his mask on, as the precautions provided for.
Reviewer comment: This does not acknowledge the bias that an experience health professional would bring to the interview, particularly preconceived ideas of the experience of those with chronic disease and how their environment will impact on their quality of life. More thought and acknowledgement of this area for potential bias is required here given to risk of bias and to ensure the robustness of the research.
Point 16: Discussion Referencing is required in the discussion. (Line 226, 228, 229, 230).
Response 16: we propose the following references
Reviewer comment: No new references are seen in the new version of the manuscript.
Reviewer comment: Section 2.8 Would still benefit from reformatting for readability, each phase does not need to be a separate paragraph.
Author Response
Point 1:
Point 10: Section 2.5 Could the observational grid be included as an example table or appendix?
Response 10: Thank you for your comment, the grid will be attached as a table
Reviewer comment: This cannot be seen in the revised manuscript. Please provide for review.
Response 1: Thank you for your vigilance, I had uploaded it along with the other pieces. I resubmit it and apologize
Point 2:
Point 11: Section 2.6 Similarly, could the interview guide be provided?Response 11: Thank you for your comment, the interview guide will be attached as a table
Reviewer comment: This cannot be seen in the revised manuscript. Please provide for review.
Response 2: Thank you for your vigilance, I had uploaded it along with the other pieces. I resubmit it and apologize
Lignes 115 et 116
Point 3:
Point 13: Line 198 What kind of training has the nurse had to prevent bias and improve neutrality during interviews and observations?
Response 13: first of all, the researcher is a trained nurse and is used to interviewing sick people. Moreover, because of COVID, the researcher kept his mask on, as the precautions provided for.
Reviewer comment: This does not acknowledge the bias that an experience health professional would bring to the interview, particularly preconceived ideas of the experience of those with chronic disease and how their environment will impact on their quality of life. More thought and acknowledgement of this area for potential bias is required here given to risk of bias and to ensure the robustness of the research.
Response 3: We thank you for your feedback and will make some changes to make the study as robust as possible
Mc Cambridge's systematic review and Hagel's study showed the influence of research participation on people's behavior. Even a covert observation would have an effect because there are ethical permissions. In order to try to minimize, the researcher will try to remain as neutral as possible by using the observation grid and the interview grid. Paradis (https://doi.org/10.1111/medu.13122) suggests the use of the phrase 'participant reactivity' when seeking to mitigate the effects of the observed person's behaviour change.
McCambridge J et al. Systematic review of the Hawthorne effect: new concepts are needed to study research participation effects. J Clin Epidemiol 2014;67:267–77
Hagel S, et al. Quantifying the Hawthorne effect in hand hygiene compliance through comparing direct observation with automated hand hygiene monitoring. Infect Control Hosp Epidemiol. 2015 Aug;36(8):957-62. doi: 10.1017/ice.2015.93
If you validate these arguments, we propose the following sentence:
Studies using covert observation can help avoid the Hawthorne effect, although even knowledge of study participation itself is considered to have the potential to induce a Hawthorne effect (27). A paragraph on the individual's reactions will be included in the final study document (28).
Point 4:
Point 16: Discussion Referencing is required in the discussion. (Line 226, 228, 229, 230).
Response 16: we propose the following references
Reviewer comment: No new references are seen in the new version of the manuscript.
Response 4:
I would like to give you the 3 references added for the discussion
- Cerezo PG, Juvé-Udina ME, Delgado-Hito P. Concepts and measures of patient empowerment: a comprehensive review. Rev Esc Enferm USP. août 2016;50(4):667‑74.
- Suárez Vázquez A, Del Río Lanza AB, Suárez Álvarez L, Vázquez Casielles R. Empower Me? Yes, Please, But in My Way: Different Patterns of Experiencing Empowerment in Patients with Chronic Conditions. Health Commun. 2017;32(7):910‑5.
- Gross O. L’empowerment, accroissement du pouvoir d’agir, est‑il éthique ? La Santé en action- Santé publique France. 2020;(453):20‑2.
Point 5:
Reviewer comment: Section 2.8 Would still benefit from reformatting for readability, each phase does not need to be a separate paragraph.
Response 5: We thank you for your comment that allows us to review the paragraph. We have removed two sentences not useful for the understanding. We have followed your suggestion in relation to the lines
The analysis will be conducted in several stages, with three researchers planned. The principal investigator will gather the data collected through observations, interviews, and informal conversations in Atlas Ti software. Initially, there will be several readings of the data and then simple coding. This step is carried out by two researchers, with a third validating in case of dispute.In the second stage, the units of meaning will be taken for a four-phase analysis according to the method proposed by Spradley (25). This method makes it possible to highlight the meaning that people give to their environment. The first phase consists of analysing the domain, i.e. grouping together the codes previously highlighted and having at least one characteristic in common, each code having a precise meaning. We thus obtain a list of domains representing behaviors in a particular context (19). The second phase is the taxonomic analysis, which will allow us to organize the codes associated with the same domain. The result is a classification of the meanings of the behaviors and the elements of the context. The third phase consists of the component analysis, which deepens both the domain analysis and the taxonomic analysis. The idea is to highlight the distinctive codes. The fourth phase is the analysis of the themes, which consists in reformulating them to make sense of people's behavior. The final analysis will highlight the subjects' strategies for managing their environment and the factors that encourage them.The validation of this analysis will also be monitored by a second researcher from the Nursing Research Chair.
